# Optimal Mobility-Aware Wireless Edge Cloud Support for the Metaverse

**Zhaohui Huang * and Vasilis Friderikos**

Center of Telecommunication Research, King's College London, London WC2R 2LS, UK
* Correspondence: zhaohui.huang@kcl.ac.uk

**Abstract:** Mobile-augmented-reality (MAR) applications extended into the metaverse could provide mixed and immersive experiences by amalgamating the virtual and physical worlds. However, the consideration of joining MAR and the metaverse requires reliable and high-quality support for foreground interactions and rich background content from these applications, which intensifies their consumption of energy, caching and computing resources. To tackle these challenges, a more flexible request assignment and resource allocation framework with more efficient processing are proposed in this paper through anchoring decomposed metaverse AR services at different edge nodes and proactively caching background metaverse region models embedded with target augmented-reality objects (AROs). Advanced terminals are also considered to further reduce service delays at an acceptable energy-consumption cost. We, then, propose and solve a joint-optimization problem which explicitly considers the balance between service delay and energy consumption under the constraints of perceived user quality in a mobility event. By also explicitly taking into account the capabilities of user terminals, the proposed optimized scheme is compared to a terminal-oblivious scheme. According to a wide set of numerical investigations, the proposed scheme has wide-ranging advantages in service latency and energy efficiency over other nominal baseline schemes which neglect the capabilities of terminals, user physical mobility, service decomposition and the inherent multimodality of the metaverse MAR service.

**Keywords:** metaverse; beyond 5G (B5G); augmented reality; mobility; structural similarity (SSIM); energy consumption



## 1. Introduction

Recently, the metaverse, which could be described as an endless virtual world where users interact with their avatars, has become popular in both academic and commercial areas [1]. Augmented reality manages to combine digital information with the real world for real-time presentation, and such experiences can be regarded as a continuum ranging from assisted reality to mixed reality, according to the level of local presence [2,3]. Mobile augmented reality (MAR), which further provides artificial perceptual information to augment the physical world during a mobility event, could be extended and enhanced in the wireless edge-supported metaverse with today's available technologies, such as digital twin and head-mounted display rendering [4,5]. In addition, compared to existing MAR applications, users could seamlessly mix their experience of the metaverse and physical world through various metaverse MAR applications, such as massively multiplayer online video games and virtual concerts [4]. Users equipped with MAR devices can upload and analyze their environment through AR customization to achieve appropriate AR objects (AROs) and access the metaverse in mobile edge networks [6]. AR marketing is regarded as having potential in metaverse, and can be seen as a typical example of a more forward-looking metaverse AR application because it replaces physical products with AR holograms and enables direct foreground interactions between the customer and the digital-marketing application interface in the background environment [2,7]. Rendering

three-dimensional (3D) AROs with the background virtual environment and updating in the metaverse consume significant amounts of energy and could be highly demanding in terms of required caching and computing resources [6,8]. Hence, such applications are delay- and energy-sensitive and face challenges in ensuring the quality of user experience and providing reliable and timely interactions with the metaverse [4,6].

Generally speaking, a metaverse scene will, in essence, consist of a background view as well as many objects in foreground interactions. The background view at a defined amalgamated virtual and physical location can be deemed as static or slowly changing [6,9]. A typical background scene can be the 3D model of the metaverse, a presentation of a related background virtual environment based on a certain user viewport [9,10]. Its size can reach tens of MB and the corresponding complexity of rendering related functionalities measured by computation load is also large (e.g., 10 CPU cycles/bit) [9,11]. On the other hand, objects (such as, for example, avatars) in foreground interactions which are embedded in the metaverse scene change much more frequently; however, they are significantly less complex than the background scene (e.g., 4 CPU cycles/bit) [8,9]. Howeve, even though those objects are less complex than the background scene, due to their frequent changes they also require rendering in a timely manner to avoid a considerable degradation in the quality of user experience. Thus, in this paper, rendering for both foreground and background are deployed at the edge clouds (ECs) rather than only at the terminals, to make full use of caching and computing resources. Notice that uploaded information is focused in foreground interactions, while background content checking consumes not only computing resources but also a significant amount of local cache to match and integrate AROs and related models of the metaverse. Hence, similar to our previous work in [12], the metaverse MAR application could also be decomposed into computational- and storage-intensive functions, which serve as a chain for improved assignment and resource allocation.

The general work flow of a metaverse MAR application supported by ECs is shown in Figure 1. A metaverse region is a fraction of the complete metaverse and is assumed to be located on a server geographically close to its corresponding service region in the mobile network (not necessarily running within an EC). The metaverse MAR service could be triggered by certain behavior including foreground interactions [6,9,13]. Then, content related to background content, such as, for example, pre-cached 3D models and AROs, are first searched in the EC cache to check if they are what the user requires. If the target AROs or model information cannot be found in the cache, then this case is labelled as a "cache miss" and the request is redirected to the original metaverse region stored in a cloud deeper in the network. Finally, according to the user's physical mobility and virtual orientation extracted from foreground interactions, the matched AROs and model are integrated into a final frame and transmitted back to the user [6,9]. At the same time, updated information is also sent to the metaverse region for synchronization. Thus, the user could be aware of changes caused by other participants if they share the same metaverse region during the service. Based on the above discussion, it is becoming apparent that the overall quality of metaverse MAR applications depends on communication delays and the capabilities of the above-mentioned network entities, which participate in service creation.

Figure 2 further reveals the difference between cases with consideration, or not, of user mobility with service decomposition to render requirements by metaverse MAR applications. Clearly, when neglecting user mobility and service decomposition, as shown in case (a), models, target AROs and metaverse MAR applications are all cached as close as possible to the user's initial location. This might leave a heavy burden for the adjacent server when there are multiple users at the same cell and cause a "hot-spot" area [12]. However, when user mobility and service decomposition are enabled, as shown in case (b), decomposed functions can spread over ECs between user's initial location and potential destination. Hence, the service delivery becomes more flexible and efficient in terms of assigning requests and allocating network resources. In case (a), although user A only needs wireless communication in the initial location, it takes two hops after moving to the middle cell. However, when taking mobility and local resources into consideration, the same user,

A, in case (b) could experience a shorter delay in the after-mobility event by allowing two more hops before the mobility event. Hence, in a high-mobility scenario, it might not always be ideal to allocate requests and services as close as possible to the user's initial location. As shown in the figure for users A and B in case (b), the AR contents in the model might be similar in terms of the viewport of different users. Hence, participating users should be aware of each other's updates and could share rendering functions to reduce the consumed resources. In this paper, we apply structural similarity (SSIM), proposed by [14], for user perception experience. It is a widely accepted method which measures the user perception quality of an image by comparing to its original version [14]. Caching more models and AROs also causes more processing and transmission delays, with energy consumption [4,6]. Hence, the joint optimization has to accept some potential loss due to constraints of computing and storage resources.

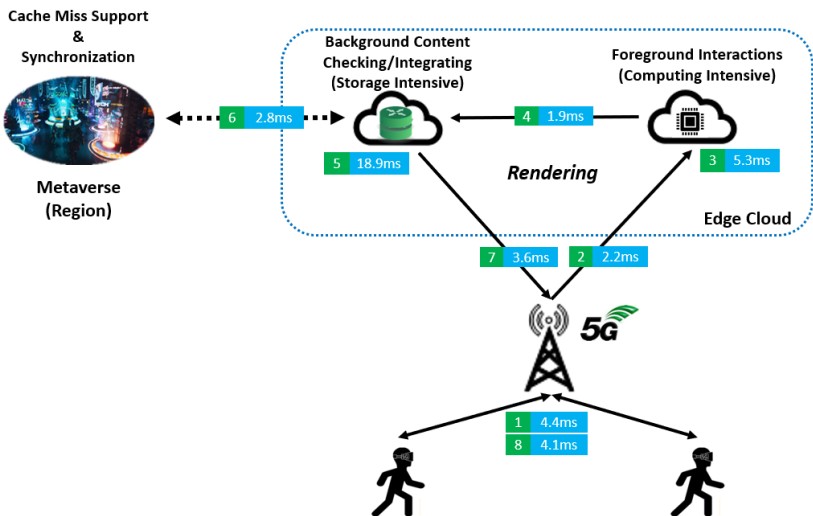

**Figure 1.** The general work flow of metaverse AR applications with delay in each stage (6 EC, 30 requests, weight $\mu$ is 1, EC Capacity is 14 and total mobility probability is 1).

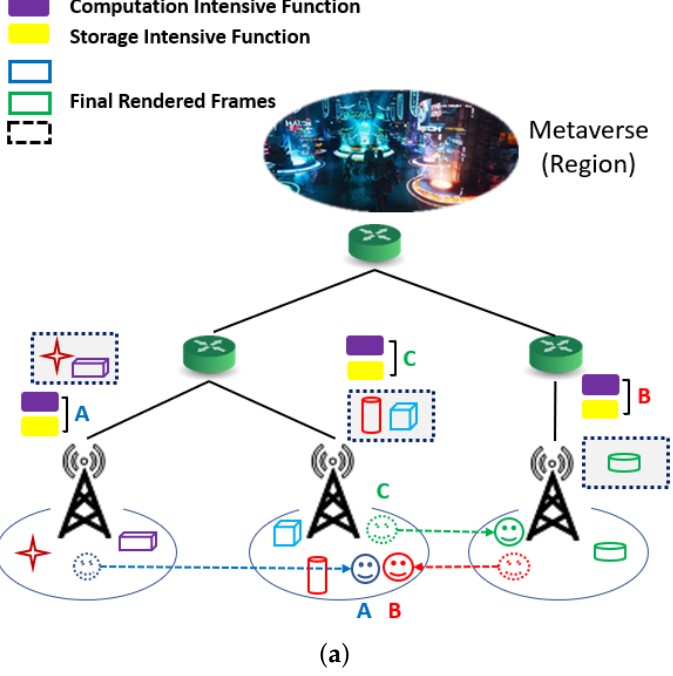

(**a**)

**Figure 2.** *Cont.*

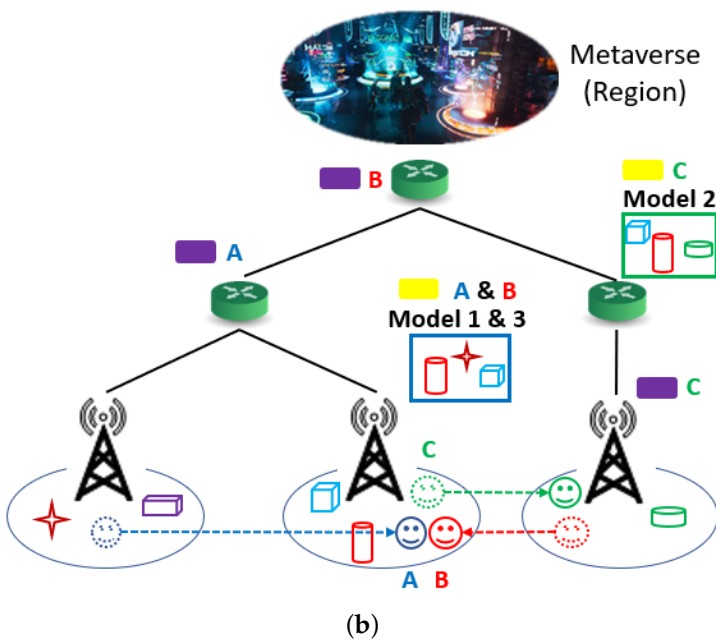

(**b**)

**Figure 2.** Illustrative toy examples of caching by a metaverse MAR application. (**a**) Traditional caching without user mobility and service decomposition. (**b**) Proactive caching with user mobility and service decomposition.

Figure 3 further reveals the difference among cases which allocate MAR service on either terminals or ECs. Note that when neglecting the EC support and service decomposition, the whole MAR application should run on the terminal and could be a heavy burden; this is shown in case (i). According to [6,15], MAR applications on terminals take up around 47% of the available power consumption and could also affect the performance of other (potentially critical) functionalities on the terminal when this service becomes more complex, such as in the metaverse. In case (ii), the processing time is significantly reduced through enabling EC support with the cost of an increased transmission delay [16]. Although the terminals are still limited in computing resources and more sensitive to energy consumption, neglecting their capabilities is not an optimal configuration, especially when recent technical improvements showcase their computing and caching potential. Noticing that the foreground interactions are much less complex than background scenes and are more suitable to terminals, we further consider terminals in the scenario, as shown by case (iii), and execute only computationally intensive functions. Hereafter, the optimization of the integration of terminals and ECs becomes the scheme OptimT and is compared to the previous case, (ii), which neglects the terminals (OptimNT). To maintain a fair comparison and focus on energy consumption and service latency, the overall energy consumption is measured explicitly (in Joules) instead of the power, as in [16], and the user perception quality is accepted as a given boundary.

In this paper, by explicitly considering the terminals, user mobility, service decomposition and models of metaverse regions with embedded AROs, a joint optimization framework (OptimT) is constructed for the metaverse MAR application in the edge-supported network. The proposed optimization framework seeks a balance between energy consumption and service delay under a given level of user perception quality. To reveal the influence of capacity and cost of terminals on the metaverse MAR application, this OptimT scheme is further compared with another optimized framework (OptimNT) proposed in our previous work [16], which only focuses on ECs and neglects terminals.

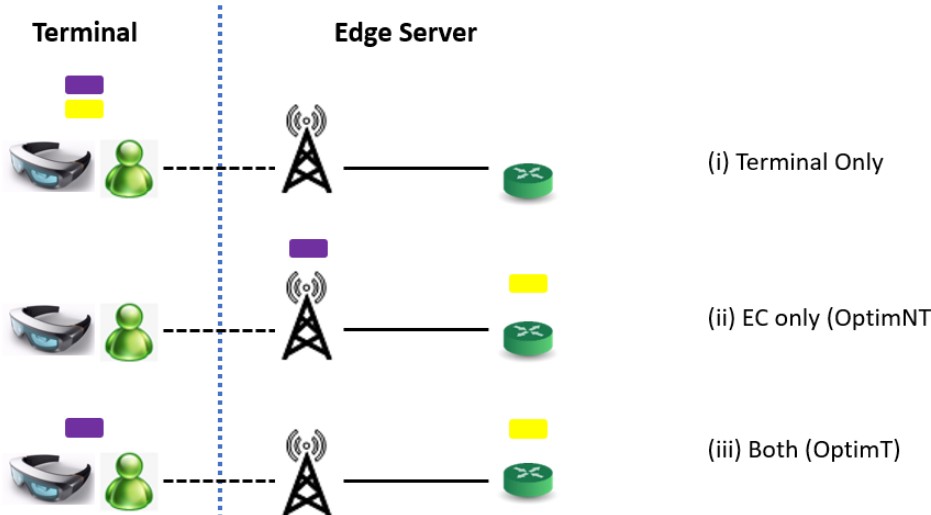

**Figure 3.** Illustrative toy examples of different activated nodes in metaverse.

## 2. Related Work

Hereafter, a series of closely related works in the area edge/cloud support of metaverse-type applications over 5G and beyond wireless networks are discussed and compared with the approach proposed in this paper.

Noticing that, despite their limitations, MAR terminals have witnessed a series of technical improvements, their joint utilization with ECs for network optimization can bring significant benefits. In [17], the authors aim to minimize the energy consumption of multicore smart devices, which are usually applied for AR applications. Through tracking the response process of an AR application, they manage to measure the terminal's energy consumption using Amdahl's law. Although the law and a similar framework for the terminal energy consumption are also applied in this paper, we consider a broader use-case scenario which includes edge servers and a more complex balance between energy and latency. The work in [18] shares a similar target to this paper, which relates to achieving a balance between latency and energy consumption under a level of acceptable image quality. However, ref. [18] brings in local sensors to MAR devices for recognizing and tracking AROs, so that their scheme can realize selective local visual tracking (optical flow) and selective image offloading. The object-recognition stage is more focused in [18], with four AR applications requiring different types of AROs. Without support from ECs, they utilize a further cloud server as a heavy database for 3D AROs and leave most tasks to the terminals, whilst cloud offloading is only triggered when confronted with a calibration. However, in this work, we consider an EC-supported network and compare the difference between schemes explicitly utilizing the MAR terminals, or not. In [19], the energy efficiency is optimized under required service latency for MAR in an EC-supported network. Similarly, the authors consider proactively caching and propose a tradeoff between energy and latency in terms of cache size. However, their mobile cache and power-management scheme still focuses the energy consumption on terminals. Clearly, the above-mentioned works do not explicitly consider user mobility, perception quality, service decomposition and metaverse application features, like ours.

The problem of efficient resource allocation for supporting metaverse-type applications is starting to attract significant amount of attention and a plethora of aspects have already been considered. In [20], the emphasis is placed on the synchronization of Internet-of-Things services, in which they employ IoT devices to collect real-world data for virtual service providers. Through calculating maximum awards, users can select the ideal virtual service provider. Researchers then propose the game framework, which considers such a reward allocation scheme and general metaverse sensing model [20]. In [21], the authors also adopt a game theoretical framework by considering tasks offloading between mobile

devices based on coded distributed computing in a proposed vehicular metaverse environment. Another framework, proposed by [22], manages and allocates different types of metaverse applications so that common resources among them can be shared through a semi-Markov decision process and an optimal admission-control scheme. The work in [23] applies a set of proposed resource-optimization schemes in a virtual education metaverse. More specifically, a stochastic optimal resource-allocation scheme is developed with the aim of reducing the overall cost incurred by a service provider. Similar to the service decomposition in this paper, they only upload and cache some parts of the data or services, to achieve reduced levels of delay and offer better privacy [23]. The work in [5] is closely related, since, in that paper, not only latency but also energy consumption is considered, as is the case for our proposed model, which uses a multi-objective optimization approach. For ultra-reliable and low-latency communication services, researchers bring in digital twins and deploy a mobility management entity for each access point, to determine probabilities of resource allocation and data offloading [5]. Then, by applying a deep-learning neuronetwork, the proposed scheme tries to identify a suitable user association and an optimized resource-allocation scheme for this association. However, in this paper, the core idea is to decompose the service and allow a flexible allocation across edge clouds by taking also into account user mobility. The work in [4] considers virtual-reality applications in the metaverse and regards the service delivery as a series of events in the market, in which users are buyers and service provides are sellers. Hence, they apply double-dutch auction to achieve a common price through asynchronous and iterative bidding stages [4]. They emphasize the quality of user perception experience by structural similarity (SSIM) and video multi-method assessment fusion. In our proposed framework, we also utilize the SSIM metric to determine the frame quality after integrating background-scene and AR contents [4]. The work in [4], further, brings in a deep reinforcement-learning-based auctioneer to reduce the information-exchange cost. While, in this paper, a multi-objective optimization approach is adopted, where we aim to balance different objective functions using the scalarization method, whilst considering the inherent user mobility in an explicit manner.

### 3. System Model

#### 3.1. Multirendering in Metaverse AR

For a given wireless network topology with the set $\mathbb{M} = \{1, 2, \ldots, M\}$, we denote the available locations, including available edge clouds and terminals. Assuming that each user makes a single request, the corresponding MAR service requests defined by $r \in \mathbb{R}$ in the metaverse region generated by mobile users could be equipped with MAR devices. Request $r$ emerges from network location $f(r)$. It represents the initial access router, to which this user is firstly connected. For each request, the user terminal could also be viewed as a valid location to cache or to process information locally. Thus, we define with $j_r \in \mathbb{M}$ the terminal sending the request $r$. The terminals are brought into consideration in the following formulation. Defining the location with constraint $j \in \mathbb{M}, j \neq j_r$ could only enable the ECs. Clearly, for the OptimNT scheme, this works for all locations. However, in the following formulation for the OptimT scheme, we force the storage-intensive functions to be executed only on the ECs, while the computing-intensive ones could also be hosted at the end terminals. During the mobility event, a user could move to different potential destinations $k \in \mathbb{K}$ (i.e., changing of the anchoring point). Hereafter, and without loss of generality, we only accept adjacent access routers as available destinations in the mobility event. A series of metaverse regions are set on ECs to interact with users. The corresponding metaverse region serving the user can be found through functions $A(f(r)), A(k)$. As explained earlier, each metaverse region is pre-deployed on a server close to the mobile network and its distance to an EC is also predefined. In this paper, as already suggested, a set of AROs is assumed to be embedded across the different background metaverse region models and is defined as $\mathbb{N} = \{1, 2, \ldots, N\}$. A set $\mathbb{S}_r = \{1, 2, \ldots, S\}$ is defined for the multiple rendering of the available metaverse region model to each user. Thus, we denote the decision variable $p_{sj}$ for pre-caching a metaverse region model $s \in \mathbb{S}_r$ at the EC $j$ ($j \in \mathbb{M}, j \neq j_r$). The subset

$\mathbb{L}_{rs}$ represents the target AROs required by the user $r$ in the related model $s \in \mathbb{S}_r$ and the size of each target ARO $l \in \mathbb{L}_{rs}$ is denoted as $O_l$. Lastly, the decision variable $h_{rl}^s$ is brought in for proactively caching an ARO required by a request, $r$. Based on the above, the decision variables $p_{sj}$ and $h_{rl}^s$ can be defined as follows

$$p^j = \begin{cases} 1, \text{ if rendering the related model } s \text{ at node } j, \\ 0, \text{ otherwise.} \end{cases} \tag{1}$$

$$h_{rl}^s = \begin{cases} 1, \text{ if ARO } l \text{ required by request } r \text{ embedded} \\ \quad \text{ in the model } s \text{ is cached,} \\ 0, \text{ otherwise.} \end{cases} \tag{2}$$

Furthermore, the additional set of constraints needs to be satisfied,

$$\sum_{r \in \mathbf{R}} h_{rl}^s \leqslant 1, \ \forall j \in \mathbf{M}, j \neq j_r, \ \forall s \in \mathbf{S_r}, \ \forall l \in \mathbf{L_{rs}} \tag{3}$$

$$\sum_{s \in \mathbf{S_r}} \sum_{l \in \mathbf{L_{rs}}} h_{rl}^s \geqslant 1, \ \forall r \in \mathbf{R} \tag{4}$$

$$\sum_{j \in \mathbf{M}, j \neq j_r} p_{sj} \geqslant h_{rl}^s, \ \forall r \in \mathbf{R}, \ \forall s \in \mathbf{S_r}, \ \forall l \in \mathbf{L_{rs}} \tag{5}$$

$$h_{rl}^s \leqslant h_{rl}^s \sum_{j \in \mathbf{M}, j \neq J_r} p_{sj}, \ \forall r \in \mathbf{R}, \ \forall s \in \mathbf{S_r}, \ \forall l \in \mathbf{L_{rs}} \tag{6}$$

Constraints in (3) force each ARO to be pre-cached at most once in a related model. Constraints (4) ensure that a valid request consists of at least one model and an embedded ARO. Constraints in (5) guarantee that the allocation of an ARO happens in conjunction with the decision to undertake proactive caching, whilst constraints in (6) further certify that the rejection of the model's proactive caching causes any ARO planned to be embedded in this model to also not be pre-cached. Thus, (5) only accepts an ARO in an pre-cached model and (6) rejects all related ones when failing to cache a model, which, together, can ensure the model and corresponding AROs cannot be handled separately during the formulation.

*3.2. Wireless Resource Allocation and Channel Model*

With $B_j$, we express the bandwidth of the resource block and $\gamma_{rj}$ denotes the signal to interference plus noise ratio (SINR) of the user $r$ at node $j$. With $P_{rj}^{tran}$, we denote the transmit power of user $r$ at node $j$, and $P_i$ is the transmission power at the base station. Furthermore, $H_{rj}$ is the channel gain, $N_j$ is the noise power and $a$ is the path loss exponent, whilst $d_{rj}$ is the distance between the user and the base station. Finally, a nominal Rayleigh fading channel is used to capture the channel between the base stations and the users [24]. More specifically, the channel gain $H_{rj}$ can be written as follows [25],

$$H_{rj} = \sqrt{\frac{1}{2}}(t + t'J) \tag{7}$$

where $J^2 = -1$, $t$ and $t'$ are random variables following the standard normal distribution. Based on the above, the SINR $\gamma_{rj}$ can be expressed as follows [25,26],

$$\gamma_{rj} = \frac{P_{rj}^{tran} H_{rj}^2 d_{rj}^{-a}}{N_j + \sum_{i \in \mathbf{M}, i \neq j} P_i H_{ri}^2 d_{ri}^{-a}} \tag{8}$$

The data rate is denoted as $g \in \mathbb{G}$ and the binary decision variable $e_{rg}$ decides whether to select the data rate $g$ for user $r$,

$$e^{rg} = \begin{cases} 1, & \text{if data rate } g \text{ is selected for user } r, \\ 0, & \text{otherwise.} \end{cases} \tag{9}$$

Noticing that the chosen data rate can also be written as $B_j \log_2(1 + \gamma_{rj})$, after choosing a data rate as $ge_{rg}$ for the user, the transmit power $P_{rj}^{tran}$ can be written as follows,

$$P_{rj}^{tran} = \frac{N_j + \sum_{i \in \mathbf{M}, i \neq j} P_i H_{ri}^2 d_{ri}^{-a}}{H_{rj}^2 d_{rj}^{-a}} \left(2^{\frac{ge_{rg}}{B_j}} - 1\right) \tag{10}$$

Note that $2^{\frac{ge_{rg}}{B_j}} = (1 - e_{rg}) + e_{rg}2^{\frac{g}{B_j}}$ and should satisfy the following constraint to ensure that a single data rate is selected per user,

$$\sum_{g \in \mathbb{G}} e_{rg} = 1, \forall r \in \mathbb{R} \tag{11}$$

### 3.3. Latency, Energy Consumption and Quality of Perception Experience

Similar to our previous work in [12], the MAR service can be decomposed into computational-intensive and storage-intensive functionalities and are defined as $\eta$ and $\varrho$, respectively. For these functionalities, their corresponding execution locations are then denoted as $x_{ri}$ and $y_{ri}$, respectively [12]. In a mobility event, the user's moving probability from the starting location to an allowable destination can be known to mobile operators through learning from the historical data and, hence, is defined as $u_{f(r)k} \in [0, 1]$ ($\{f(r), k\} \subset \mathbb{M}$). The size of foreground interactions is denoted as $F_{\eta r}^{fore}$, the size of pointers used for matching AROs is denoted as $F_{\varrho r}$, and the size of the related model $s$ used for background content checking is $F_{sr}^{back}$ [9,12]. During the matching and background-content validation process, the target AROs and/or the background content are possibly not pre-cached in the local cache, and such a case is known as a "cache miss" (otherwise there is a "cache hit"). A cache miss in the local cache inevitably triggers the redirection of the request to the metaverse region stored in a core cloud deeper in the network and this extra cost in latency is defined as the penalty $D$. After rendering, the model and target AROs are integrated into a compressed final frame for transmission and its compressed size is denoted as $F_{sr}^{res}$.

In this section, a joint optimization scheme is proposed which aims to balance the service delay and the energy consumption under the constraint of the user perception quality of the decomposed metaverse AR services in the EC supported network. The cache hit/miss is expressed by the decision variable $z_{rj}$ and can be written as follows,

$$z_{rj} = \begin{cases} 1, & \text{if } \sum_{l \in \mathbf{L_{rs}}} \sum_{s \in \mathbf{S_r}} p_{sj} h_{rl}^s \geqslant L_{rs}, \\ 0, & \text{otherwise.} \end{cases} \tag{12}$$

The cache capacity of an EC and the cache hit/miss relation can be written as follows,

$$\sum_{r \in \mathbf{R}} \sum_{l \in \mathbf{L_{rs}}} \sum_{s \in \mathbf{S_r}} p_{sj} h_{rl}^s O_l \leq \Theta_j, \forall j \in \mathbf{M}, \ j \neq j_r \tag{13}$$

$$\sum_{l \in \mathbf{N}} \sum_{s \in \mathbf{S_r}} h_{rl}^s + \epsilon \leq L_{rs} + U(1 - q_{rj}) \tag{14}$$

$$\forall j \in \mathbf{M}, \ j \neq j_r, r \in \mathbf{R}$$

where $\Theta_j$ denotes the cache available memory at node $j$. In (14), to transfer the either-or constraint (i.e., $\sum_{l \in \mathbf{N}} \sum_{s \in \mathbf{S_r}} h_{rl}^s < L_{rs}$ or $z_{rj} = 1$) into inequality equations, we bring in $\epsilon$ as a small tolerance value, $U$ as a large arbitrary number and $q_{rj}$ as a new decision variable satisfying $1 - q_{rj} = z_{rj}$ [12]. Undoubtedly, increased levels of pro-caching decisions related to the background models and embedded AROs in a request inevitably brings about an extra execution burden for the matching function. Taking the above into account, the actual processing delay of the computational-intensive function can be expressed as follows,

$$V_{rj} = \frac{\omega_\eta F_{\eta r}^{fore}}{f_V^j} \tag{15}$$

Similarly, the processing delay of the matching and background-content checking function is assumed to happen only at servers, and can be written as

$$W_{rj} = \frac{\omega_\varrho (F_{\varrho r} + \sum_{l \in \mathbf{L_{rs}}} \sum_{s \in \mathbf{S_r}} p_{sj} h_{rl}^s O_l + \sum_{s \in \mathbf{S_r}} F_{sr}^{back} p_{sj})}{f_V^j} \tag{16}$$

where $\omega_\eta$ and $\omega_\varrho$ (cycles/bit) represent the computation load of foreground interaction and background matching, $f_V^j$ is the virtual CPU frequency (cycles/sec), and $F_{\varrho r}$ is the size of uploaded pointers of AROs in foreground interactions [9,12]. When finding the target AROs during matching, their pointers, included in foreground interactions, should also be transferred to the metaverse for updating. Finally, the final frame integrating the model and target AROs are transmitted back to the user. Hence, the overall transmission delay for each user after processing using the functions can be written as

$$\sum_{s \in \mathbf{S_r}} \sum_{j \in \mathbf{M}, j \neq j_r} (C_{jA(f(r))} + C_{jA(k)}) p_{sj} + \\ (C_{A(f(r))f(r)} + \sum_{k \in \mathbf{K}} C_{A(k)k} u_{f(r)k}) \tag{17}$$

Note that the product of decision variables, $p_{sj} h_{rl}^s$, $p_{sj} y_{rj}$ and $p_{sj} h_{rl}^s y_{rj}$, creates a non-linearity. Observe that $p_{sj} h_{rl}^s$ and $p_{sj} y_{rj}$ appear directly while the product $p_{sj} h_{rl}^s y_{rj}$ appears in $W_{rj} y_{rj}$, which represents the execution of the matching function at the location $j$ ($j \in \mathbb{M}, j \neq j_r$). To express the optimization problem in a nominal linear programming setting, we linearize the above expressions via new auxiliary decision variables. To this end, a decision variable $\alpha_{rsj}$ is introduced as $\alpha_{rsj} = p_{sj} y_{rj}$ and the constraints should be added, as follows

$$\alpha_{rsj} \leqslant p_{sj}, \\ \alpha_{rsj} \leqslant y_{rj}, \\ \alpha_{rsj} \geqslant p_{sj} + y_{rj} - 1 \tag{18}$$

Similarly, a new decision variable $\beta_{rslj}$ is introduced as $\beta_{rslj} = p_{sj} h_{rl}^s$ and the constraints should be added as follows

$$\beta_{rslj} \leqslant p_{sj}, \\ \beta_{rslj} \leqslant h_{rl}^s, \\ \beta_{rslj} \geqslant p_{sj} + h_{rl}^s - 1 \tag{19}$$

The constraints in (6) can be rewritten as follows,

$$h_{rl}^s \leqslant \sum_{j \in \mathbf{M}} \beta_{rslj}, \ \forall r \in \mathbf{R}, \ \forall s \in \mathbf{S_r}, \ \forall l \in \mathbf{L_{rs}} \tag{20}$$

In addition, it is worth pointing out that for the decision variable $p_{sj}$, the following holds: $p_{sj} = p_{sj}^2$. Therefore, we have $p_{sj} h_{rl}^s y_{rj} = \alpha_{rsj} \beta_{rslj}$. Hence, a decision variable $\lambda_{rslj}$ is defined as $\lambda_{rslj} = \alpha_{rsj} \beta_{rslj}$ and the following set of constraints are added

$$
\begin{aligned}
\lambda_{rslj} &\leqslant \alpha_{rsj}, \\
\lambda_{rslj} &\leqslant \beta_{rslj}, \\
\lambda_{rslj} &\geqslant \alpha_{rsj} + \beta_{rslj} - 1
\end{aligned}
\tag{21}
$$

Hence, the product $W_{rj} y_{rj}$ can be rewritten as follows,

$$
\frac{\omega_\varrho \left( F_{\varrho r} y_{rj} + \sum_{l \in \mathbf{L_{rs}}} \sum_{s \in \mathbf{S_r}} \lambda_{rslj} O_l + \sum_{s \in \mathbf{S_r}} F_{sr}^{back} \alpha_{rsj} \right)}{f_V^j}
\tag{22}
$$

By checking whether users share the same metaverse region by $A(f(t)) = A(f(r)), \{t, r\} \subset \mathbb{R}$, we can ensure the user could also view other updates happening in the same metaverse region. Based on the previous modelling of the wireless channel, the wireless transmission delay in a mobility event can be written as follows,

$$
\begin{aligned}
\sum_{r \in \mathbf{R}} &\frac{F_{\eta r}^{fore} + \sum_{t \in \mathbf{R}, A(f(t)) = A(f(r))} \sum_{s \in \mathbf{S_r}} p_{sj} F_{st}^{res}}{g e_{rg}} + \\
\sum_{r \in \mathbf{R}} \sum_{k \in \mathbf{K}} u_{f(r)k} &\frac{F_{\eta r}^{fore} + \sum_{t \in \mathbf{R}, A(t) = A(k)} \sum_{s \in \mathbf{S_r}} p_{sj} F_{st}^{res}}{g e_{rg}}
\end{aligned}
\tag{23}
$$

Noticing that with constraint (11), $\frac{1}{e_{rg}}$ can be replaced with $e_{rg}$ for linearization by introducing a new decision variable $\phi_{rlsg}$ with the following constraints,

$$
\begin{aligned}
\phi_{rsg} &\leqslant e_{rg}, \\
\phi_{rsg} &\leqslant p_{sj}, \\
\phi_{rsg} &\geqslant e_{rg} + p_{sj} - 1
\end{aligned}
\tag{24}
$$

Thus, the previous Formula (23) can be updated as follows,

$$
\begin{aligned}
\frac{1}{g} \sum_{r \in \mathbf{R}} \left( 1 + \sum_{k \in \mathbf{K}} u_{f(r)k} \right) \Big( F_{\eta r}^{fore} e_{rg} + \\
\sum_{t \in \mathbf{R}, A(f(t)) = A(f(r))} \sum_{s \in \mathbf{S_r}} \phi_{rsg} F_{st}^{res} \Big)
\end{aligned}
\tag{25}
$$

Based on the above derivations and in-line with [12], the overall latency can be written as follows,

$$
\begin{aligned}
L = (25) + \sum_{r \in \mathbf{R}} \sum_{i \in \mathbf{M}} (C_{f(r)i} + V_{ri}) x_{ri} + \\
\sum_{r \in \mathbf{R}} \sum_{i \in \mathbf{M}} \sum_{j \in \mathbf{M}, j \neq j_r} ((22) + C_{ij} \xi_{rij} + C_{A(f(r))f(r)} + \psi_{rj} D) + \\
\sum_{s \in \mathbf{S_r}} \sum_{j \in \mathbf{M}, j \neq j_r} (C_{jA(f(r))} + C_{jA(k)}) p_{sj} + \\
\sum_{r \in \mathbf{R}} \sum_{k \in \mathbf{K}} (C_{A(k)k} + C_{ki} x_{ri}) u_{f(r)k}
\end{aligned}
\tag{26}
$$

where $V_{ri}$ is the processing delay of a computational-intensive function [12]. $L_{max}$, here, denotes the maximum allowed service latency and has $\frac{L}{L_{max}} \in [0, 1]$.

The energy efficiency of the system during each service time slot is measured by the production of its total power and running time. The server total power consists of the transmission power and CPU processing power at target ECs. Denote the required CPU processing power of the user $r$ at the node $j$ as $P_{rj}^{cpu}$ and the CPU-chip architecture coefficient as $k_0$ (e.g., $10^{-18}$) [5]. Then, the power at the EC can be achieved through $k_0(f_V^j)^2$ (J/s), based on measurements in [27,28]. Noticing that for both conditions, the background contents are processed at servers, the consumed processing time of the server is

$$T_{cpu} = \sum_{r \in \mathbf{R}} \sum_{j \in \mathbf{M}, j \neq m_r} V_{rj} x_{rj} + \sum_{r \in \mathbf{R}} \sum_{j \in \mathbf{M}} W_{rj} y_{rj} \tag{27}$$

The wireless transmission happens regardless of whether foreground interactions are being executed at terminals. Since the selected data rate is $ge_{rg}$, the wireless transmission time is

$$T_{tran} = \sum_{r \in \mathbf{R}} \sum_{j \in \mathbf{M}, j \neq j_r} \frac{F_{\eta r}^{fore}}{ge_{rg}} + \sum_{r \in \mathbf{R}} \sum_{j \in \mathbf{M}, j = j_r} \frac{F_{\varrho r}}{ge_{rg}} \tag{28}$$

Finally, the total consumed energy at the server side can be written as follows,

$$
\begin{aligned}
E_{server} &= \sum_{r \in \mathbf{R}} \sum_{j \in \mathbf{M}} (P_{rj}^{tran} T_{tran} + P_{rj}^{cpu} T_{cpu}) \\
&= \sum_{r \in \mathbf{R}} \sum_{j \in \mathbf{M}} (\frac{N_j + \sum_{i \in \mathbf{M}, i \neq j} P_i H_{ri}^2 d_{ri}^{-a}}{H_{rj}^2 d_{rj}^{-a}} (2^{\frac{ge_{rg}}{B_j}} - 1)) \\
&\quad (\sum_{j \neq j_r} \frac{F_{\eta r}^{fore}}{ge_{rg}} + \sum_{j = m_r} \frac{F_{\varrho r}}{ge_{rg}}) \\
&\quad + \sum_{r \in \mathbf{R}} \sum_{j \in \mathbf{M}} k_0(f_V^j)^2 (W_{rj} y_{rj} + \sum_{j \neq j_r} V_{rj} x_{rj})
\end{aligned}
\tag{29}
$$

Regarding the terminal side, in this paper, we follow the Amdahl's law to model the energy consumption, in which the potential speedup of potential parallel computations is considered in the function [29]. In this paper, the metaverse AR functions work by serial and, for simplicity, the parallel portion is assumed to be zero like in [17]. As mentioned earlier, the dynamic foreground interactions, including some highly used AROs, could be proactively cached at terminals with the matching function. The 3D background model, on the other hand, is much larger and might serve multiple users in a region. Hence, it is not recommended to store or process this at the terminal. In addition, the metaverse application should not exceed a certain portion of the whole terminal CPU resources, so that other functionalities can work properly [17]. Denoting the consumed portion as $\Gamma_r \in [30\%, 50\%]$ [17], then, $\frac{V_{rj} x_{rj}}{\Gamma_r}$ means processing metaverse AR functions at the terminal requires a longer time. The energy consumption of terminals can be written as follows,

$$
\begin{aligned}
E_{terminal} &= P_{terminal} T_{terminal} \\
&= \sum_{r \in \mathbf{R}} \sum_{j = j_r} k_0(f_V^j)^2 \frac{V_{rj} x_{rj}}{\Gamma_r}
\end{aligned}
\tag{30}
$$

Finally, the overall system energy consumption is $E = E_{server} + E_{terminal}$. $E_{max}$ represents the maximum possible energy consumption of the system. It also has $\frac{E}{E_{max}} \in [0, 1]$.

SSIM is applied to reveal the quality of perception experience. In this paper, the video coding scheme (e.g., H.264) and frame resolution (e.g., $1280 \times 720$) are assumed as pre-defined [10]. Then, SSIM is mainly affected by data rate and a concave function can be applied to reveal the relation between them [10]. Hence, the set of SSIM values for each

ARO under corresponding data rates can be denoted as $\mathbb{SSIM}_l, l \in \mathbb{L}_r$. The overall quality of perception experience, $Q$, can be written as follows,

$$Q = \sum_{r \in \mathbf{R}} \sum_{l \in \mathbf{L_r}} \sum_{g \in \mathbf{G}} \sum_{c \in \mathbf{SSIM}_l} e_{rg} c \tag{31}$$

To maintain the user experience above an acceptable level, the perception-quality constraint can be added, as follows,

$$\frac{Q}{Q_{max}} \geq Q_{bound} \tag{32}$$

where $Q_{max}$ is the maximum quality through selecting max allowable data rate and storing as many AROs as possible.

Using a weighting parameter $\mu \in [0, 1]$, the bi-opbjective optimization problem can be written as follows,

$$min \ \mu \frac{L}{L_{max}} + (1 - \mu) \frac{E}{E_{max}} \tag{33a}$$

$$\text{s.t. } z_{rj} = 1 - q_{rj}, \ \forall j \in \mathbf{M}, r \in \mathbf{R} \tag{33b}$$

$$\sum_{r \in \mathbf{R}} (x_{rj} + y_{rj}) \leq \Delta_j, \forall j \in \mathbf{M}, j \neq j_r \tag{33c}$$

$$\sum_{j \in \mathbf{M}} x_{rj} = 1, \forall r \in \mathbf{R} \tag{33d}$$

$$\sum_{j \in \mathbf{M}, j \neq j_r} y_{rj} = 1, \forall r \in \mathbf{R} \tag{33e}$$

$$\xi_{rij} \leq x_{ri}, \ \forall r \in \mathbf{R}, i, j \in \mathbf{M}, j \neq j_r \tag{33f}$$

$$\xi_{rij} \leq y_{rj}, \ \forall r \in \mathbf{R}, i, j \in \mathbf{M}, j \neq j_r \tag{33g}$$

$$\xi_{rij} \geq x_{ri} + y_{rj} - 1, \ \forall r \in \mathbf{R}, i, j \in \mathbf{M}, j \neq j_r \tag{33h}$$

$$\psi_{rj} \leq z_{rj}, \ \forall r \in \mathbf{R}, j \in \mathbf{M}, j \neq j_r \tag{33i}$$

$$\psi_{rj} \leq y_{rj}, \ \forall r \in \mathbf{R}, j \in \mathbf{M}, j \neq j_r \tag{33j}$$

$$\psi_{rj} \geq z_{rj} + y_{rj} - 1, \ \forall r \in \mathbf{R}, j \in \mathbf{M}, j \neq j_r \tag{33k}$$

$$x_{rj}, y_{rj}, p_{sj}, h_{rl}^s, z_{rj}, q_j \in \{0, 1\},$$

$$\alpha_{rsj}, \beta_{rslj}, \lambda_{rslj}, \phi_{rslg}, \psi_{rj}, \xi_{rij} \in \{0, 1\},$$

$$\forall r \in \mathbf{R}, j \in \mathbf{M}, l \in \mathbf{L_{rs}}, s \in \mathbf{S_r} \tag{33l}$$

$$(3), (4), (5), (20), (11), (13), (14),$$

$$(18), (19), (21), (24), (32)$$

As mentioned earlier, any assignment relating to the storage-intensive functions ($y_{rj}$) is limited to only ECs and, hence, should apply the constraint $j \neq j_r$. The constraint (33b), together with constraints (3) to (20), express the interrelationship between the pre-caching decisions and the cache miss/hit for each request [12]. The constraint (33c) reflect the limitation of a virtual machine, whilst constraints (33d) and (33e) guarantee that each function should only be executed once at a single server [12]. Finally, the constraints (18) to (21) and (33f) to (33k) relate to the auxiliary variables that have been used for linearization.

## 4. Numerical Investigations

In this section, the effectiveness of the proposed optimization scheme, which is referred to as Optim in the following, is investigated and is compared with a number of nominal (baseline) mechanisms.

A nominal tree-like network topology, as shown in Figure 4, was applied with 20 ECs in total, 6 ECs being activated for the current metaverse AR service and 30 requests being sent by MAR devices. The remaining available resources allocated for metaverse AR support within an EC were assumed to be CPUs with frequencies of 4 to 8 GHz, CPU-chip architecture coefficient of $10^{-18}$ (affected by the chip's design and structure) [5], 4 to 8 cores and $[100, 400]$ MBytes of cache memory [12]. Similarly, the mobiles AR devices were assumed to have a CPU with 1 GHz frequency, 4 cores and $[0, 100] MBytes$ available cache memory for metaverse AR applications [17]. According to [15], the power of AR applications should not exceed 50% of the mobile device's CPU total power (2–3 W), so that other functionalities can operate efficiently. Hereafter, a nominal frame rate of 15 frames/second is assumed and the rendering takes place every other frame (˜133.2 ms interval) [30–32]. Thus, the service delay of the aforementioned work flow within the above time interval could be regarded as acceptable. Each request requires a single free resource unit for each service function, such as, for example, a virtual machine (VM) [33]. Up to 14 available VMs are assumed in each EC, with equal splitting of the available CPU resources [12]. Note that different view ports lead to different models of the metaverse [9] and up to four different models can be cached. All target AROs must be integrated into the corresponding model and rendered within the frame before being streamed to the end user based on a matched result. After triggering the metaverse MAR service, pointers to identify AROs such as a name or index are usually a few bytes [34] and, hence, their transmission and processing are neglected in the following simulations. The set of available data rates is $\{2, 3, \ldots, 8\}$ Mbps and its corresponding SSIM values set are $\{0.955, 0.968, \ldots, 0.991\}$ [10]. We require the acceptable average SSIM above 0.97 ($Q_{bound}$). For a nominal 5 G base station, we assume its cell radius to be 250 m, its carrier frequency 2 GHz, its transmit power 20 dBm, the noise power $10^{-11}$ W, the path loss exponent to be 4, its maximum resource blocks 100 and, without the loss of generality, each user can utilize only one resource block [35–37]. As mentioned earlier, we accept a predefined video coding scheme H.264 with a fixed frame resolution of $1280 \times 720$ [10] in RGB (8 bits per pixel). Based on the given resolution, the size of foreground interactions after decoding and compressing can be calculated through multiplying the coefficients $\frac{5}{9}$ and $10^{-3}$ [9]. Matlab on a personal PC with a CPU of intel i7, 6500U and 2 cores was employed for the simulation. Key simulation parameters are shown below, in Table 1.

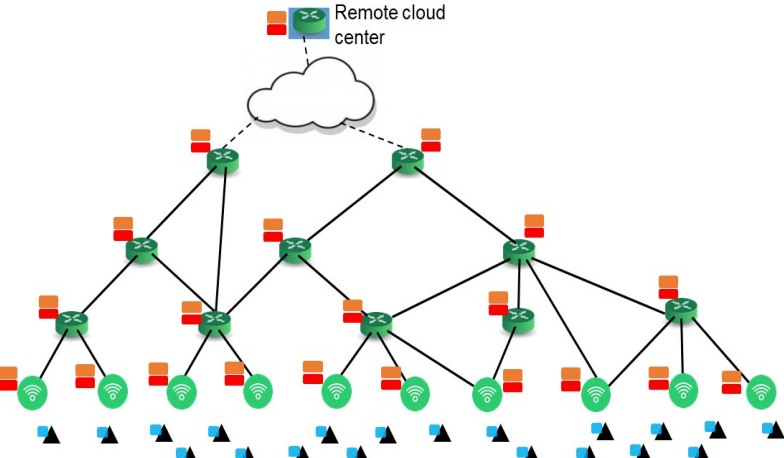

**Figure 4.** A typical tree-like designed network topology.

In the following figures and discussion, the optimized scheme considering terminals is denoted as OptimT, while the other onem which does not include the terminals, is denoted as as OptimNT. The OptimNT scheme could be regarded as a natural extension of our previous work [16]. Two other baseline schemes sharing same caching decisions as the proposed Optim scheme were also implemented for comparison. Those are the random

selection scheme (RandS) and the closest EC first scheme (CEC) [38]. The RandS scheme operates a random EC selection while the other two both select the closest EC to the user's initial location. The CEC scheme also accepts the second closest EC as a back up choice [38].

**Table 1.** Simulation parameters.

| Parameter | Value |
| --- | --- |
| Number of available ECs | 6 |
| Number of available VMs per EC (EC capacity) | 14 |
| Number of requests | 30 |
| Number of available models per user | 4 |
| AR object size | $(0, 10]$ MByte |
| Total moving probability | $[0, 1]$ |
| Cell radius | 250 m |
| Remained cache capacity per EC | $[100, 400]$ MByte |
| EC CPU frequency | [4,8] GHz |
| EC CPU cores | [4,8] |
| EC CPU core portion per VM | 0.25–0.5 |
| Remained cache capacity per terminal | $[0, 100]$ MByte |
| Terminal CPU frequency | 1 GHz |
| Terminal CPU cores | 4 |
| CPU architecture coefficient | $10^{-18}$ |
| Foreground-interaction computational load | 4 cycles/bit |
| Background-content-checking computational load | 10 cycles/bit |
| Carrier frequency | 2 GHz |
| Transmission power | 20 dBm |
| Path loss exponent | 4 |
| Noise power | $10^{-11}$ W |
| Number of resource blocks | 100 |
| Frame resolution | $1280 \times 720$ |
| Average latency per hop | 2 ms |
| Cache miss penalty | 25 ms |

According to Figure 5, the service delay for each request of the proposed schemes decreases, as expected, with an increasing weight $\mu$. With a larger weight, the proposed schemes tend to select a larger data rate and direct the service to more powerful ECs, which naturally leads to a smaller overall delay. Compared to the OptimNT scheme, for example, the gain in delay of the OptimT scheme ranges from 1.9% to 10.4%. When seeking the best energy efficiency ($\mu = 0$), since the CPU resources at the terminals are also shared by other functionalities, the OptimT scheme also tries to avoid the occupation of terminals. Hence, in this case, these two schemes share similar solutions and approaches. Noticing that the proposed schemes do not care about latency cost, they could choose a further away and busy EC, which even causes the OptimNT scheme to perform worse than the CEC scheme. Afterwards, as the weight $\mu$ increases and the emphasis is placed on latency rather than energy, the OptimT scheme allocates some foreground interactions to terminals and becomes better than OptimNT. Since the baseline schemes neglect energy consumption, service decomposition and mobility, their gaps in relation to the proposed scheme become

larger with increasing weight. However, such a gain in delay comes with an extra cost in energy consumption. As shown in Figure 6, the energy consumption per request of the proposed scheme increases with a larger weight. Compared to the OptimNT scheme, the OptimT scheme consumes 2.9% to 23.0% more energy under different weights. Thus, it might not always be worth enduring lots of energy consumption for narrow gains in delay. By selecting a suitable weight, a balance could be achieved with the OptimT scheme between delay and energy consumption. Through the utilization of terminals, the OptimT scheme becomes the most sensitive to energy consumption and there could be instances in which it might consume more energy than the RandS scheme.

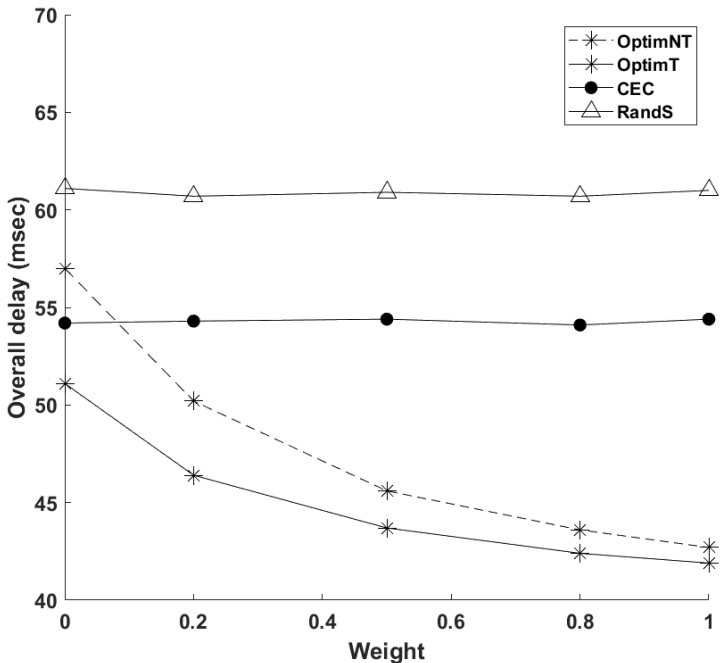

**Figure 5.** Overall delay with weight $\mu$ (6 EC, 30 requests, EC capacity is 14 and total mobility probability is 1).

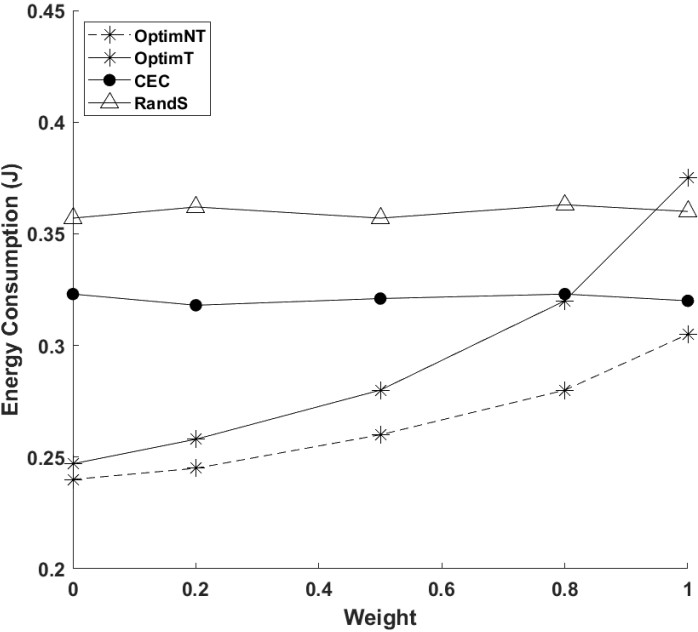

**Figure 6.** Average energy consumption with weight $\mu$.

Clearly, the user experience could be elevated through viewing more AROs in foreground interactions or more delicate scenes from background models. Figure 7 reveals the variation in delay with the increasing foreground-interaction size. When the average foreground interaction size is not too large and there are still enough resources at target ECs, OptimNT and baseline schemes increase almost linearly at a similar speed. When the size keeps increasing and resources become limited, the CEC scheme becomes the most sensitive because it only targets the several closest ECs and it is easier to trigger the penalty. The OptimT scheme, on the other hand, maintains the least latency and the least increasing tendency. To this end, it could save up to 13.8% and 51.7% delay compared, respectively, to the OptimNT scheme and the CEC scheme. Figure 8 further reveals the variation in energy in this case. Baseline schemes process foreground interactions at ECs without considering energy. Hence, their decisions are not obviously affected by the size of foreground interactions and their energy consumption increases almost linearly. The OptimNT scheme keeps finding more suitable ECs according to current foreground-interaction size and remaining resources, while the OptimT schemes further enables terminals to process some tasks. Compared to the CEC scheme, they could save over 14.3% energy. It is necessary to point out that the energy consumption will not be taken into account when redirecting the request to the farther core cloud and triggering the penalty. According to Figure 9, the background model size is much larger and could cause a significant increase in delay. Note that the terminals could take charge of some foreground interactions and, hence, could make room for background models in the OptimT scheme. It is still the best scheme in terms of delay, which could be up to 9.8% less than the OptimNT scheme. As mentioned earlier, the proactive caching and processing of background models only happen at ECs; these schemes share a similar level of increased energy consumption until triggering the overloading penalty.

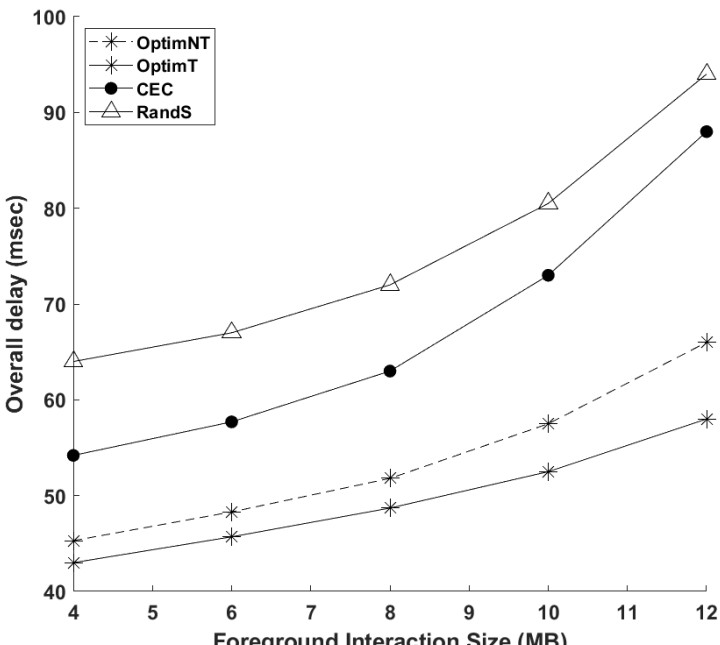

**Figure 7.** Overall delay with foreground interaction size (6 EC, 30 requests, $\mu = 0.5$, EC capacity is 14 and total mobility probability is 1).

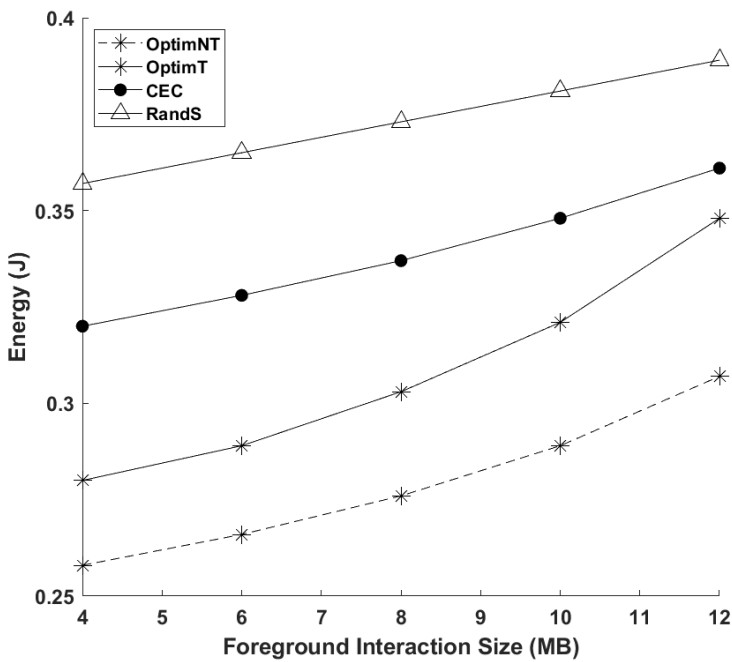

**Figure 8.** Average energy consumption with foreground interaction size (6 EC, 30 requests, $\mu = 0.5$, EC capacity is 14 and total mobility probability is 1).

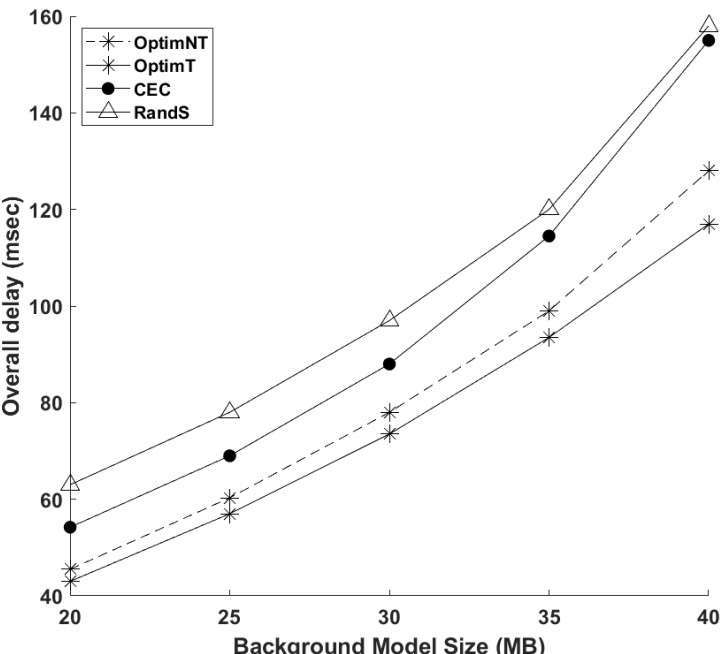

**Figure 9.** Overall delay with background model size (6 EC, 30 requests, $\mu = 0.5$, EC capacity is 14 and total mobility probability is 1).

The number of available VMs activated in an EC is known as the EC capacity. For a given EC capacity (e.g., 14), the ratio between the different number of requests (e.g., [30,40]) and the EC capacity could be applied to represent the average EC utilization in the network. Then, this rate was normalized into $[0, 1]$ for better presentation. As shown in Figures 10 and 11, the increasing EC utilization rate indicates a more congested network and, hence, as expected, the delay and energy consumption increase as well. Compared to the OptimT scheme, the OptimNT scheme is still more sensitive in terms of energy but better in terms of delay. Thus, its consideration of terminals benefits delay at the cost of energy. Observe from Table 2, that even when there is no mobility event, the proposed OptimT

scheme is still slightly better than other baseline schemes because its flexibility of terminals can better avoid potential EC overloading. Therefore, the proposed OptimT and OptimNT schemes have an obvious advantage over baseline schemes and are recommended in a congested network and a high-user-physical-mobility scenario. Especially when the MAR terminal still has enough energy capacity, its computing resources should not be neglected and, hence, the OptimT scheme is more suitable in this case.

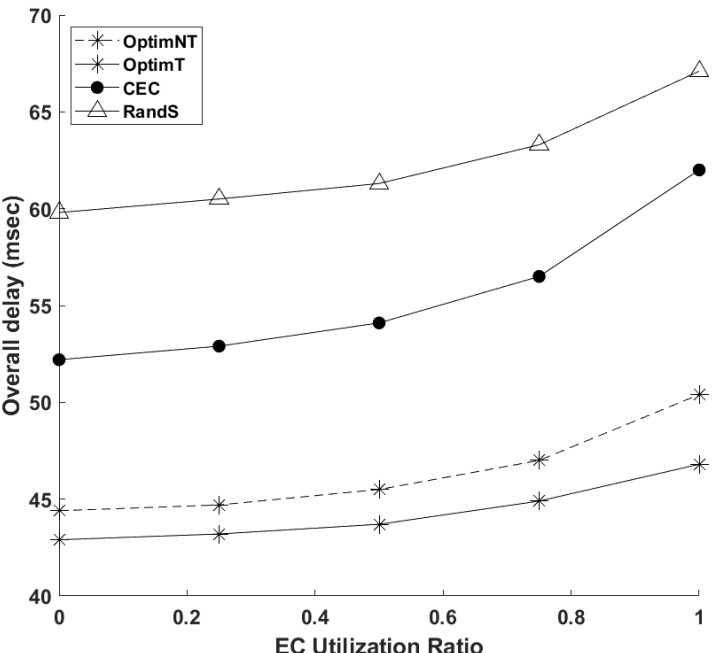

**Figure 10.** Overall delay with average EC utilization rate (6 EC, $\mu = 0.5$ and total mobility probability is 1).

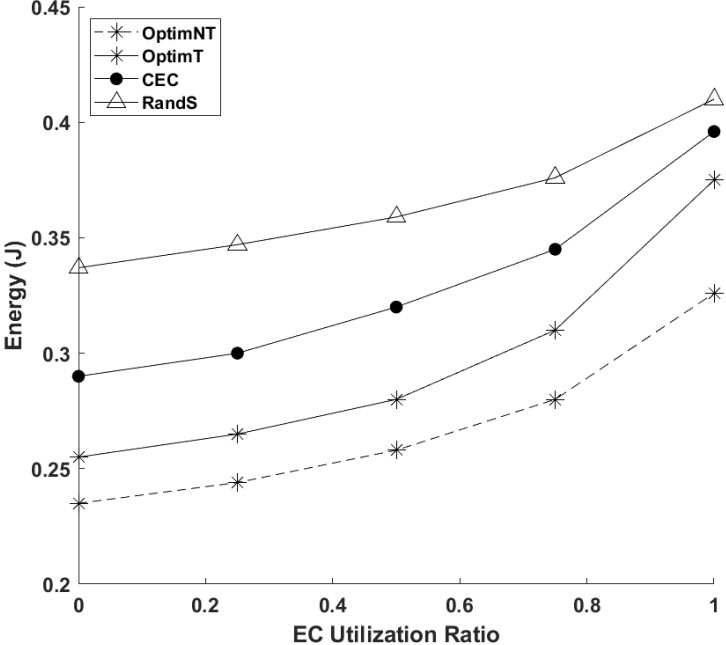

**Figure 11.** Energy consumption with average EC utilization rate (6 EC, $\mu = 0.5$ and total mobility probability is 1).

**Table 2.** Overall delay in no-mobility event ($\mu = 1$, 6 ECs, 30 requests and EC capacity is 14).

| Scheme | OptimT | OptimNT | CFS | RandS |
| --- | --- | --- | --- | --- |
| Delay (ms) | 38.8 | 40.1 | 40.7 | 60.8 |

## 5. Conclusions

Extending MAR applications into the metaverse is expected to incorporate the rendering and updating of high-quality AR metadata in order to provide a more realistic experience. Hence, such forward-looking applications are highly delay- and energy-sensitive and are significantly demanding in terms of caching and computing resources. In this paper, a joint optimization scheme is proposed by explicitly considering the model rendering performance, user mobility and service decomposition to achieve a balance between energy consumption and service delay under the constraint of user perception quality for metaverse MAR applications. Recent technical improvements in AR devices allow them to process more tasks locally. To this end, we explore their potential in the metaverse and compare the performance with terminal-oblivious schemes which reside on cloud support. A wide range of numerical investigations reveals that the proposed terminal-aware framework provides improved decision making compared to baseline schemes for energy consumption and resource allocation for metaverse MAR applications, especially under a congested network and high-mobility scenario.

**Author Contributions:** Z.H., writing—original draft, writing—review and editing; V.F., writing—review and editing, supervision. All authors have read and agreed to the published version of the manuscript.

**Funding:** This research received no external funding.

**Data Availability Statement:** Necessary data are included in this paper.

**Conflicts of Interest:** The authors declare no conflict of interest.

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
