# Peer review of "Optimal Mobility-Aware Wireless Edge Cloud Support for the Metaverse"

_futureinternet, doi:10.3390/fi15020047_

Round 1
Reviewer 1 Report
The manuscript “Optimal Mobility Aware Wireless Edge Cloud Support for the Metaverse” is interesting and focuses on an area not so common. The authors put much effort into creating this paper, and the results can be important for many disciplines. The paper is well-structured, and the findings are clearly presented. References are enough and up-to-date. I just have a few comments.
The authors present a sophisticated approach in several steps. The first part of the article (Introduction) presents the difference between cases that consider the user mobility with service decomposition or not related to rendering requirements in metaverse AR (figure 2). This issue is not clearly described in the text. There is a lack of connection between the text and the symbols used in the figure. The current version of the description does not fully reflect the difference between the two approaches. Please consider improving this part of the article.
Some specific comments, including adding and modifying figures, are listed below.
· Please place the Figures and Tables immediately after they are first mentioned so that the paragraphs are not splitted, and the reading is eased. Currently, in the text, figure 6 is before figure 5.
· There is a lack of consistency in naming. For example – sometimes authors used “optimT” and then “OptimT.” It would be good if the authors revised and improved the text.
· All abbreviations should be defined in parentheses the first time they appear in the text and used consistently after that. Due to the many abbreviations used in the text, I would suggest considering adding a table containing all abbreviations used in the paper.
· Figures 2 and 3 – “Illustrative toy examples of …” please consider changing the figure caption. In its current form, the caption is not intuitive and hardly intelligible.
The discussion and conclusions are presented clearly.
For the above reasons, I would consider publishing this paper after minor revision.
Author Response
Thanks for your comments. Our reponse to each reviewer and the revised version paper are attached below.

Reviewer 2 Report
The paper is on mobile augmented reality applications in the metaverse. It deals with the problem of optimizing resource allocation with respect to latency and energy consumption by explicitly considering terminals, user mobility, service decomposition, etc.
This is an important problem. The proposed solution and research methodology in the paper are sound and the simulation results appear to support the conclusion.
The following are some minor comments:
- Abstract: Typo "=".
- The use of uppercase characters in abbreviations is inconsistent.
- The writing needs to be improved. Past and present tenses are often mixed.
- To improve readability, I'd suggest organizing long paragraphs into shorter paragraphs.
- Details on the hardware/software used for the simulation should be elaborated.
Author Response

(The authors gave the same response as above.)

Reviewer 3 Report
Here is my anonymous review to be shared with the authors only.
Overall, interesting work, yet needs some conceptual clarity:
1) define metaverse according to Matthew Ball (2022)
2) define XR and AR according to the xr framework (http://tiny.cc/ydvzuz )
3) There is work that defines Augmented Reality Marketing incl. "reduced reality" and studies that look at how AR holograms can substitute physical products. Such workd should be part of the discussion especially since such use cases help readers to better understand AR Metaverse
4) A strength of your paper is that you use AR and Metaverse. Most people have use Metaverse as a synonym for social VR. Please highlight this strength better. So comment 3 as an example of how this can be achieved
5) Please use a professional copy editor. The paper needs to be streamlined.
5) I can't evaluate the technical part, needs to be done by another reviewer.
6) Please reduce the numbers of abbreviations and use terms/abbreviations consistently (e.g. MAR vs. on p. 19 "Extending mobile Augmented Reality applications into the metaverse")
7) Please discuss hardware and the role of "local presence". Smartphone are nice for today's AR and they are good to study and learn about the metaverse; but for a true metaverse, we need handsfree devices. Check the work on "Augmented Reality Smart Glasses" (ARSGs)
Author Response

(The authors gave the same response as above.)

Round 2
Reviewer 3 Report
Obverall, the paper has improved. The definition of AR is still missing. Please fix. Reference as suggested last time. Besides that, ready for publication.
----
open review: No
publication of review: No
Author Response
Thanks a lot. We add the definition of AR in this version at the start of Introduction before going into Mobile AR. This does make the intro look more reasonable.
